# Predicted third-order sweet spots for $\phi_0$-junction Josephson parametric amplifiers

Tasnum Reza and Sergey M. Frolov

*Department of Physics and Astronomy, University of Pittsburgh, Pittsburgh, PA, 15260, USA*

(Dated: March 26, 2025)

Hybrid superconductor-semiconductor nanowire Josephson junctions exhibit skewed and $\phi_0$-shifted current phase relations when an in-plane magnetic field is applied along the weak link's spin-orbit effective field direction. These junctions can have an asymmetric Josephson potential with odd-order nonlinearities. A dominant third-order nonlinearity can be achieved by tuning the magnetic field to a sweet spot. Sweet spots persist when higher order Josephson harmonics are included. This makes it possible to have a single Josephson junction dipole element with three-wave mixing capability, which is favorable for pump-efficient amplification. Electrostatic gate tunability of the semiconductor weak link can make it operable within an extended range of working frequencies, and the inclusion of micromagnets can facilitate near-zero magnetic field operation.

## I. INTRODUCTION

Quantum limited parametric amplifiers are key components for low-noise microwave signal amplification in qubit readout circuits [1–3]. The most common parametric amplifiers are based on the so-called fourth order nonlinear term of the inductive potential, also known as the Kerr term. This term provides for robust and practical operation. However, it leads to pump-dependent frequency shifts, constrains the pump frequency to values close to the signal frequency, and also causes gain saturation at higher signal powers, limiting the dynamic range of amplification [4, 5].

Parametric amplifiers that employ predominantly third-order nonlinearity in the Josephson potential [6–8] show higher dynamic range and avoid frequency shifts [9–11]. A common device with third-order nonlinearity is the Superconducting Nonlinear Asymmetric Inductive eLement (SNAIL) [6, 12]. Besides the three-wave mixing quantum-limited parametric amplification, SNAILs have other applications such as parametric coupling/conversion, and gate operations in the so-called cat qubits [13–17].

In typical microwave parametric amplifiers, the nonlinear inductance of a a tunnel Josephson junction is used to build an anharmonic oscillator. The current-phase relation (CPR) characterizes the nonlinearity of a Josephson junction; for tunnel junctions CPR is sinusoidal: $I(\phi) = I_0 \sin(\phi)$ where $I(\phi)$ is the Josephson supercurrent, $\phi$ is the phase difference between the superconducting leads, and $I_0$ is the critical current. Expanding to the lowest order of nonlinearity ($\phi \ll 1$) gives the fourth order term in the Josephson potential which dictates the microwave dynamics of nonlinear circuit elements in devices such as qubits and parametric amplifiers [1, 2, 18–22]. Typically, multi-junction circuits are employed to engineer SNAILs.

More generally, the CPRs of mesoscopic Josephson junctions can contain a sum of higher harmonic modes of the sine function, i.e, $I(\phi) = \sum_n I_{n \geq 1} \sin(n\phi)$ [23]. For instance, in the one-dimensional short junction limit, the Josephson energy is given by $U(\phi) =$ $-\Delta \sum_i \sqrt{1 - \tau_i \sin^2(\phi/2)}$, where $\tau_i$ are the transmissions for the multiple conduction channels in the semiconductor and $\Delta$ is the induced superconducting gap [24, 25]. While hybrid nanowire Josephson junctions (JJs) such as Sn/InSb or Sn/InAs are likely not in the short junction limit, their CPRs are nevertheless expected to have higher order harmonics [26].

The presence of higher-order CPR terms in itself is not sufficient to realize a single junction dominated by third-order nonlinearities, but a skewed $\phi_0$-junction phenomenon offers that advantage [27–29]. It refers to a Josephson junction with a CPR that lacks symmetry upon inversion of either the current or the phase. The effect can emerge under spatial inversion and time-reversal symmetry breaking. In nanowires with strong spin-orbit interaction, it appears when a magnetic field is applied along the direction of the effective spin-orbit field. The bias-asymmetric critical currents, which may or may not indicate a skewed $\phi_0$-junction, were reported as "superconducting diode" effects in multiple studies [30–41].

We propose that a skewed $\phi_0$-junction can be used to realize a SNAIL-like device based on a single Josephson junction [42]. The inductive potential of a nanowire $\phi_0$-junction can be optimized to zero-out fourth-order Kerr terms. We find the Kerr-free sweet spots for two-harmonic and three-harmonic CPRs derived from tight-binding simulations, and compared with experiments.

By way of assessment of device characteristics from key metrics such as power gain, 1 dB compression power and operable frequency bandwidth, Josephson junctions based on Sn as superconductor offer the critical current levels (100s of nA) that are compatible with SNAIL operation. The device requires large magnetic fields in order to break time-reversal symmetry. Such fields are present by default in certain quantum applications such as quantum dot-based spin qubits and putative toplogical qubits. For near-zero field use, fixed local fields can be generated by micromagnets. Single nanowire parametric amplifiers can potentially be more compact and simpler to control. Additionally, gate-voltage tunability of nanowire junctions facilitates control over a large bandwidth of working frequency (on the order of a few GHz).

arXiv:2503.19891v1 [cond-mat.mes-hall] 25 Mar 2025

## II. SKEWED $\phi_0$-JUNCTION

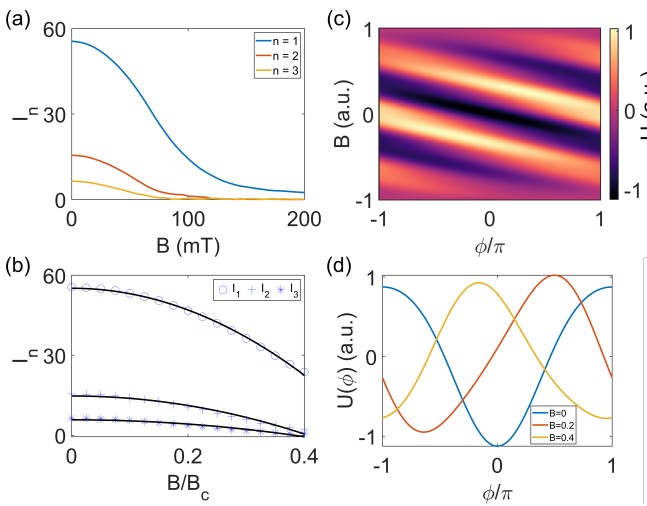

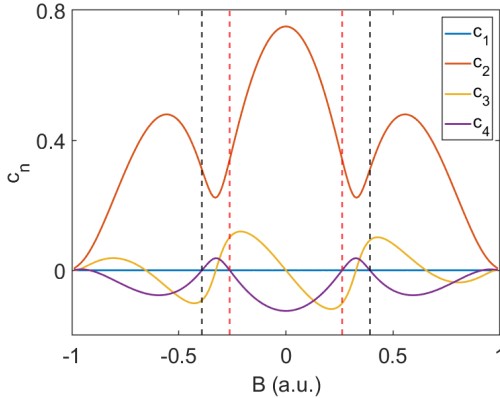

FIG. 2. Non-linear coefficients $c_n$ as function of $B$ for a set of optimized values $a = 8.83$ and $c = 9.60$. Sweet spots with $|c_4| \propto 0$ and a significant $|c_3|$ are at $B \approx \pm 0.26, \pm 0.39$ shown by red and black dashed lines.

FIG. 1. (a) Amplitudes $I_n$ from numerical simulations as a function of field $B$. (b) Numerical data (symbols) and fits (solid lines) for $I_n = \eta_{n1}(1 - \eta_{n2}B^2)$, where $\eta_{n1}$ and $\eta_{n2} \approx 1$ are fitting parameters. (c) Skewed $\phi_0$-junction potential as function of field $B$ and phase $\phi$ for optimized values $a = 8.83$ and $c = 9.60$. (d) Line-cuts of potential $U(\phi)$ from panel (c).

We first illustrate the skewed $\phi_0$-junction current-phase relation by postulating it empirically in the following minimal form:

$$I(\phi) = I_1 \sin(\phi + \phi_0) + I_2 \sin(2\phi + 2\phi_0 + \delta_{12}) \quad (1)$$

This function lacks symmetry upon inversion of both $\phi$ and $I$, allowing for large third-order nonlinearities in the Josephson potential. In quasi one-dimensional weak links, we assume that amplitudes $I_1$ and $I_2$, as well as phase shifts $\phi_0$ and $\delta_{12}$ are dependent on magnetic field $B$ applied along the spin orbit field direction:

$$\begin{aligned} I(\phi) = &\alpha(1 - B^2)\sin(\phi + aB) \\ &+ \beta(1 - B^2)\sin(2\phi + 2aB + cB) \end{aligned} \quad (2)$$

where $\phi_0 = aB$, $\delta_{12} = cB$, $I_1 = \alpha(1-B^2)$, $I_2 = \beta(1-B^2)$, for $B << B_c$, where $B_c = 1$ is the critical field. $a$, $c$, $\alpha$ and $\beta$ are constants. The Josephson potential is:

$$\begin{aligned} U(\phi) = &- \alpha(1 - B^2)\cos(\phi + aB) \\ &- \frac{\beta}{2}(1 - B^2)\cos(2\phi + 2aB + cB) \end{aligned} \quad (3)$$

This CPR is an approximation of a numerical CPR from a realistic tight-binding model simulation performed using KWANT [43, 44], which in turn aims to reproduce experiments in Ref. [45]. The Hamiltonian used for this model is a 3D hybrid superconductor-semiconductor nanowire with spin-orbit interaction in a magnetic field. Using the data generated by the simulation, we fit the behavior of $I_1$, $I_2$, $\phi_0$ and $\delta_{12}$ as a function of $B$ (Fig. 1).

For the amplitudes of the Josephson harmonics $I_n$ we use values from the Fourier expansion of the numerical current-phase relation [46]. In Fig. 1(a) we plot $I_n$ vs $B$ from simulation data. In Fig. 1(b), we show the fit for the first three harmonics of the CPR. The fit functions closely follow the postulated quadratic relation. We also find that phases $\phi_0$ and $\delta_{12}$ increase linearly with field within the range $\sim 0 - 100$ mT, here we set $B_c = 200$mT for fitting purposes. We find the ratio $I_1/I_2 \sim 4$. We perform our analysis in the next section using the first two Josephson harmonics, and we expand to a third in the Appendix B.

The potential function $U(\phi)$ in Eq. (3) is asymmetric in $\phi$ as shown Fig. 1(d), and thus it generates both even and odd order terms upon expansion. The parameters $a$ and $c$ are chosen to be in the single-well potential regime. Taylor-expanding $U(\phi)$ at the minimum of the potential well ($\tilde{\phi} = \phi - \phi_{\min}$) gives the coefficients for nonlinear interaction terms $c_n$:

$$U(\tilde{\phi}) = c_2\tilde{\phi}^2 + c_3\tilde{\phi}^3 + c_4\tilde{\phi}^4 + ... \quad (4)$$

We impose $c_1 = 0$ at the potential minimum. The quadratic term in Eq. (4) defines a simple harmonic oscillator. All higher-order terms contribute to the anharmonicity. Next, we show that by varying the magnetic field the device can be tuned to a sweet spot regime where the $c_3$ nonlinearity is dominant, and $c_4 = 0$.

## III. THREE-WAVE MIXING SWEET SPOTS

We follow a potential optimization strategy similar to that used for conventional SNAILs [6, 10, 12] to search for a Kerr-free three-wave mixing regime. We explore the

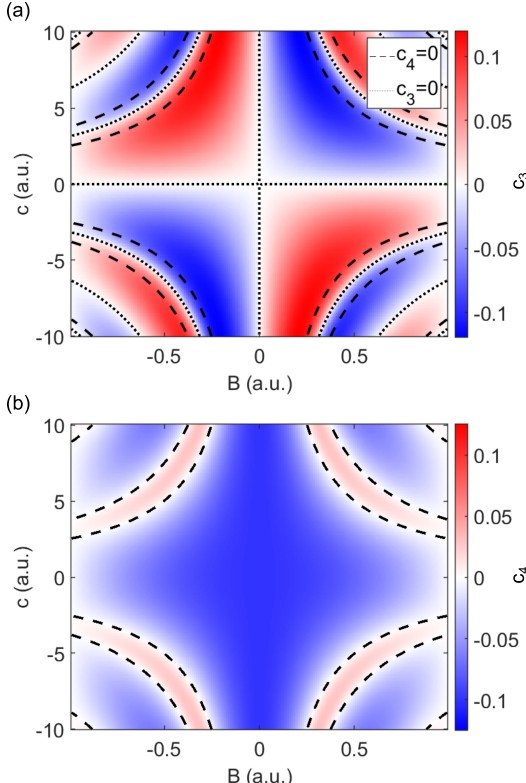

FIG. 3. (a) Nonlinear coefficient $c_3$ as function of $c$ and $B$ at $a = 1$. Black dashed lines at $c_4 = 0$, black dotted lines at $c_3 = 0$. (b) Nonlinear coefficient $c_4$ as a function of $c$ and $B$ at $a = 1$. Black dashed lines at $c_4 = 0$.

space of $a$ and $c$ which parametrize how phase shifts $\phi_0$ and $\delta_{12}$ evolve with $B$. We use an optimizing algorithm (hybrid genetic algorithm implemented in MATLAB [47]) to find points that minimize $|c_4|$ and maximize $|c_3|$. Fig. 2 shows that in the optimized regime we arrive at sweet spots for particular magnetic fields.

The sweet spots are not exclusive to these specific values of $a$ and $c$. In fact, there can be multiple such points on a pareto frontier. This increases the likelihood of reaching these settings in real nanowire Josephson junctions where CPR can vary from device to device. This is illustrated in Fig. 3, where we fix $a$ and vary $c$ while monitoring $c_3$ and $c_4$. The condition for $c_4 = 0$ and $c_3 \neq 0$ is met over a large parameter space. Also, varying $a$ for a fixed $c$ does not change $c_3$ or $c_4$ because $a$ parametrizes the global phase shift $\phi_0$, this is shown in the Appendix A.

In the above calculations, we consider only the first two harmonics of the CPR. However, the dynamics in a real device may be influenced by the presence of higher-order harmonic terms. In Appendix B, we show how adding a third order harmonic in the CPR would affect the nonlinear behavior of the device. While the behavior is more complex, the Kerr-free sweet spots are still present.

## IV. DEVICE PROPOSAL

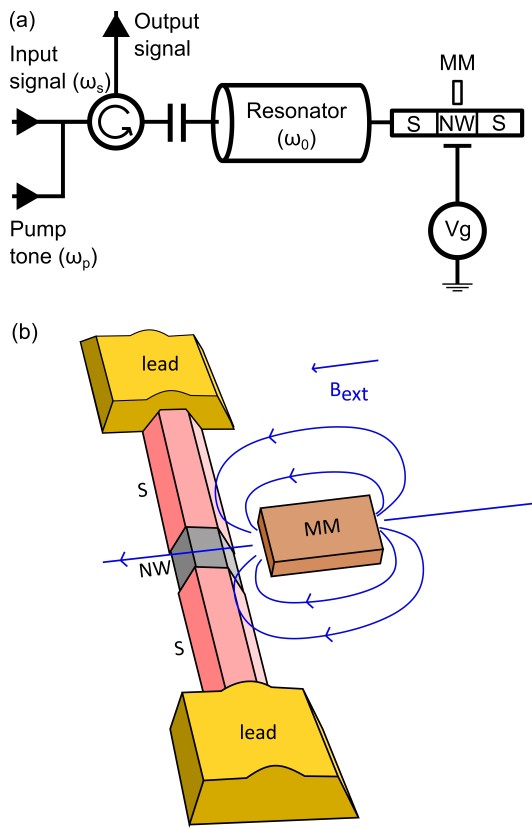

FIG. 4. (a) Schematic of a parametric amplifier circuit with nanowire (NW) Josephson junction coupled to a resonator. A micromagnet (MM) is placed close to the junction and NW electron density is tuned with a gate voltage $V_g$. (b) Detailed schematic of the NW and MM with thin superconducting shell (S), a break in the shell defines the JJ. An external magnetic field $B_{\text{ext}}$ can be used in conjunction with the micromagnet to fine-tune the effective field near the sweet spot.

The $\phi_0$-junction Josephson parametric amplifier device consists of a hybrid superconductor-semiconductor nanowire junction as the key nonlinear component incorporated into a resonator and coupled to a transmission line (Fig. 4(a)). As discussed above, the nonlinearity can be tuned with a magnetic field applied in the effective spin-orbit field direction, typically perpendicular to the nanowire, to induce the skewed and asymmetric CPR effect. If zero-field operation is of interest, we propose using stray fields from a micromagnet placed in proximity to the nanowire (Fig. 4(b)). In previous studies [48–51] micromagnets made of materials such as CoFe were used to generate strong local fields (upto$\sim$ $\pm100$ mT) near the nanowire junction. Junctions with thin superconducting shells have high in-plane critical fields of the order of $\sim 1$ T [52]. A small parallel field can also be applied to fine-tune the total magnetic field to a Kerr-free sweet spot. The field strength and direction can be designed using micromagnetics simulations platforms such

as MuMax3 [53]. By placing the large superconducting shapes a distance away from the micromagnet, the global circuit including the resonator can be unaffected by local magnetic field effects: we estimate microTelsa stray perpendicular fields at distances of tens of microns away from the micromagnet.

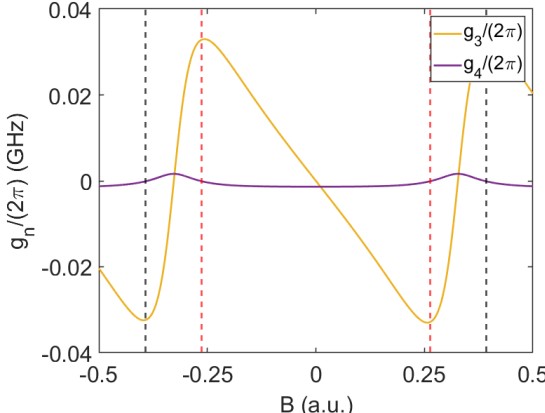

FIG. 5. Nonlinear terms $g_3$ and $g_4$ as a function of field $B$. Black and red dashed lines show sweet points where $|g_3| > 0$ and $g_4 = 0$. Here $\omega_0 = 20$ GHz, $L_J \approx 3.2$ nH and $L \approx 0.4$ nH.

In order to estimate the behavior of the $\phi_0$-junction in a microwave circuit, we calculate the experimentally relevant terms $g_3$ and $g_4$ for the SNAIL Hamiltonian that includes the JJ and the resonator. Under rotating wave approximation and $\phi \ll 1$ (detailed in Appendix C), the Hamiltonian is:

$$\hat{H}/\hbar = \omega_r \hat{a}^\dagger \hat{a} + g_3(\hat{a} + \hat{a}^\dagger)^3 + g_4(\hat{a} + \hat{a}^\dagger)^4 \quad (5)$$

where $\omega_0$ is the bare resonator frequency, $L_J$ is the Josephson junction inductance and $L$ is the linear inductance of the circuit. The characteristic frequency $\omega_r = \frac{\omega_0}{\sqrt{(1+L_J/c_2 L)}}$ [10] should fall within a few GHz frequency range typical for standard microwave experiments as shown in Fig. 11. For Sn-InSb and Sn-InAs hybrid nanowire junctions, critical current at zero field is of order $100 - 500$ nA, tunable with gate voltage. Therefore, the Josephson energy $E_J \sim 50 - 250$ GHz. Choosing realistic circuit parameters we apply the magnetic field evolution discussed in Sec. II to estimate $g_3$ and $g_4$ (Fig. 5).

While the power gain of such a device will be experimentally limited by a variety of factors, the upper bound on it can be estimated from the input-output theory using the nonlinear Hamiltonian in Eq. (5) (truncated at $n = 4$) [10]:

$$G = 1 + \frac{4\kappa^2 |g_{\text{eff}}|^2}{(\Delta_{\text{eff}}^2 - \omega^2 + \frac{\kappa^2}{4} - 4|g_{\text{eff}}|^2)^2 + (\kappa\omega)^2} \quad (6)$$

where $\omega = \omega_s - \omega_p/2$, $g_{\text{eff}} = 2g_3\alpha_p$, $\Delta_{\text{eff}} = \Delta + 12g_4(\frac{8}{9}|\alpha_p|^2 + |\alpha_s|^2 + |\alpha_i|^2)$, $\Delta = \omega_r - \omega_p/2$. $\omega_p$, $\omega_s$ and $\omega_i$ are the pump, signal and idler frequencies ($\omega_p = \omega_s + \omega_i$). $\kappa$ is the dissipation rate due to the coupling to the transmission line and $\alpha_s$, $\alpha_p$ and $\alpha_i$ are the intra-cavity amplitudes of the pump, signal, and idler.

Following Fig. 5, at the Kerr-free point $|g_3|/2\pi \approx 33$ MHz and $\omega_r/2\pi \approx 3.98$ GHz. The term $g_4$ is negligible ($|g_4|/2\pi \approx 12$ Hz limited by the numerical resolution). Assuming $|\alpha_s| \approx |\alpha_i| = 0.01$ for low signal power and taking the coupling rate $\kappa/2\pi = 0.4$ GHz, we find a gain of $G_0 = 20$ dB that can be achieved over a dynamic bandwidth of 40 MHz, shown in the Appendix D.

The input signal power for which the gain drops by 1dB, $P_{-1\text{dB}}$, is typically dependent on $|g_4|$ for fourth-order amplifiers. In SNAILs, $g_3$ as well as other factors can lead to gain saturation [10]. For optimal performance $g_3$ may need to be reduced in addition to tuning $\kappa$ and the energy participation ratio [54].

## V. CONCLUSIONS

In conclusion, we propose a single-junction Josephson parametric amplifier based on a hybrid superconductor-semiconductor nanowire operating in the three-way mixing mode. Specifically, we study the skewed $\phi_0$-junction current-phase relation, which exhibits an asymmetric potential with odd-order nonlinearities. This effect is induced when an in-plane magnetic field is applied along the effective spin orbit field direction.

We find sweet spots where third-order nonlinearity dominates while the fourth-order nonlinearity is suppressed, an ideal condition for implementing the three-wave mixing scheme. We show that these sweet spots are present across a wide range of parameters of the two-component CPR. The sweet spots persist when a third-order Josephson harmonic is added to the CPR. We estimate the device performance to meet the standards of state-of-the-art quantum-limited parametric amplifiers. We also predict that the electrostatic gate tuning of the NW JJ will enable a wide range of operational frequencies for amplification and potential three-wave mixing. We propose a micromagnet-based design for near-zero field operation. The single-junction approach can help miniaturize three-wave Josephson parametric amplifiers which sometimes employ $\sim$20 SNAIL arrays [10]. Future work will focus on the experimental implementation of this device, including the design and the optimization of the microwave circuit implemented using magnetic-field compatible superconducting materials.

## VI. CODE AND DATA AVAILABILITY

All codes and data are available on Zenodo [47].

## VII. DURATION OF STUDY

This project was started in 2023, with about a year of active work.

## VIII. ACKNOWLEDGMENTS

We thank M. Hatridge for useful discussions. This work is supported by the U.S. Department of Energy under grant DE-SC-0019274.

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

**Appendix A: Dependence on a global phase shift $\phi_0$**

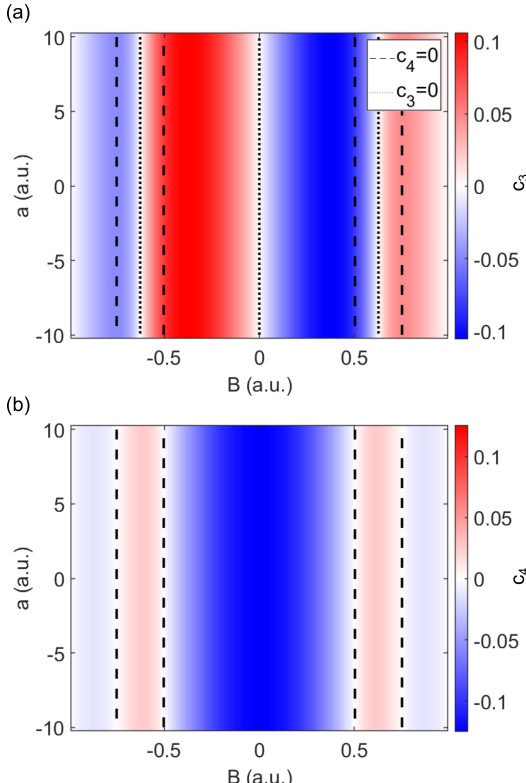

FIG. 6. (a) Nonlinear coefficient $c_3$ as a function of $a$ and $B$ at $c = 5$. Black dashed lines at $c_4 = 0$, black dotted lines at $c_3 = 0$. (b) Nonlinear coefficient $c_4$ as a function of $a$ and $B$ at $c = 1$. Black dashed lines at $c_4 = 0$.

As shown in Fig. 6, $c_3$ and $c_4$ do not depend on $a$, i.e. they are independent of a global phase shift $\phi_0$.

**Appendix B: Sweet spots with higher order CPR harmonics**

Adding a third order harmonic term in the CPR, we can rewrite it as:

$$I(\phi) = I_1 \sin(\phi + \phi_0) + I_2 \sin(2\phi + 2\phi_0 + \delta_{12}) \\ + I_3 \sin(3\phi + 3\phi_0 + \delta_{13}) \quad \text{(B1)}$$

where $I_1, I_2, I_3 \propto (1 - B^2)$ and $\phi_0, \delta_{12}, \delta_{13} \propto B$. We use the relations $\phi_0 = aB$, $\delta_{12} = cB$, $\delta_{13} = dB$ and $B_c = 1$. From the KWANT simulation data fit Fig. 1, we arrive at $I_2/I_1 \approx 1/4$ and $I_3/I_1 \approx 1/9$. Following the same optimization process, with the goal to maximize $|c_3|$ and constrain $|c_4| = 0$, we find Kerr-free sweet points, shown in Fig. 7.

We also look at how the addition of the third harmonic phase shift $\delta_{13}$ influences the characteristic behavior of $c_n$. We show $c_n$ as a function of $d$ and $B$ at $a = 1$,

$c = 5$ in Fig. 8. As the relative strength of the third harmonic component $I_3$ is small, the potential landscape is not changed significantly. However, in the special case of $I_2$ and $I_3$ being dominantly large, it may perturb the behavior as discussed in the Appendix B.1.

Finally, we also look at the higher order nonlinear terms and search for points where all nonlinear terms $c_n \approx 0$ except for $c_5$ and $c_6$, shown in Fig. 9. In this case, at $a = 1$, $c = 20$ we find points where $c_6$ is the dominant term at $B \sim \pm 0.6$, $d \sim \pm 5$. More such points exist at higher values of $c$ and across a range of values of $d$. It has been suggested in the literature that the higher order nonlinear interactions can be useful for hardware-efficient quantum error correction schemes [42, 55].

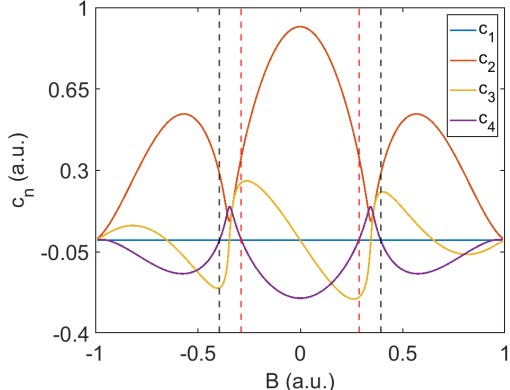

FIG. 7. Non-linear coefficients $c_n$ as function of $B$ when adding a third order harmonic term to the CPR. Sweet spots $|c_4| \propto 0$ and $|c_3| > 0$ are at $B \approx \pm 0.4, \pm 0.3$ shown by red and black dotted lines. $a = 7.16$, $c = 8.64$, $d = 9.95$.

**B.1. CPR with large second and third harmonic components**

When the second and third harmonics of the CPR are large, e.g. $I_1/I_2 = 1/2$ and $I_1/I_3 = 1/3$ (a possibility for more exotic Josephson junctions), we find that a double-well potential. A discontinuity occurs in the calculated nonlinear coefficients at the degeneracy point when the state switches from one minimum to the other (shown in Fig. 10). In the case of multiple local minima, we choose the global minimum for calculations.

**Appendix C: Nonlinear Hamiltonian**

We start with the Hamiltonian as follows:

$$H = 4E_C N^2 + U(\tilde{\phi}) \quad \text{(C1)}$$

where $E_C$ is the total charging energy of the device, $N$ is the number of Cooper pairs, and $E_J$ is the Josephson

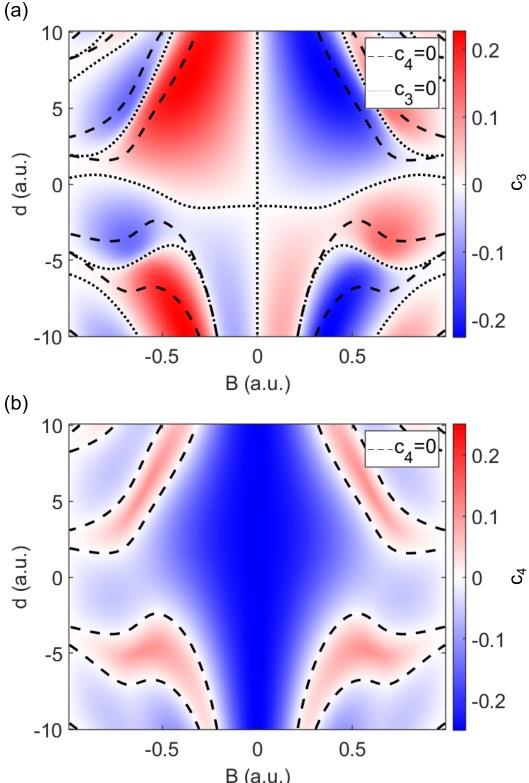

FIG. 8. (a) Nonlinear coefficient $c_3$ as a function of $d$ and $B$ at $a = 1$, $c = 5$, and (b) Nonlinear coefficient $c_4$ as a function of $d$ and $B$ at $a = 1$, $c = 5$, when adding a third order harmonic in CPR.

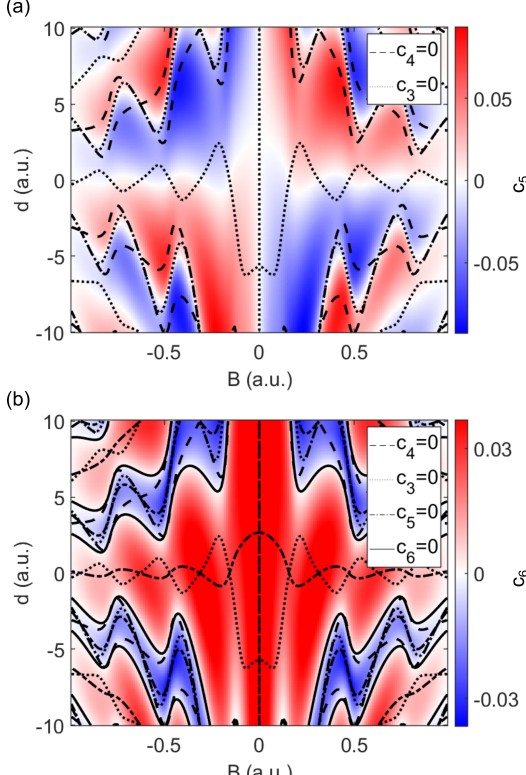

FIG. 9. (a) Nonlinear coefficient $c_5$ as a function of $d$ and $B$ at $a = 1$, $c = 20$, and (b) Nonlinear coefficient $c_6$ as a function of $d$ and $B$ at $a = 1$, $c = 20$, when adding a third order harmonic in CPR

charging energy. We can add a shunted capacitance C to control total $E_C$ and $E_J/E_C$ according to the circuit physics at play. $E_C \ll E_J$ in the regime where device is insensitive to charge noise.

The device can be treated as an essentially weakly nonlinear harmonic oscillator given by the Hamiltonian:

$$H = H_0 + H_{\text{nl}} \tag{C2}$$

Where $H_0$ is the linear part and $H_{\text{nl}}$ is the nonlinear part of the Hamiltonian.

Upon Hamilton quantization, where the phase operator is $\hat{\phi} = \phi_{\text{zpf}}(\hat{a}^\dagger + \hat{a})$ and the number operator is $\hat{N} = iN_{\text{zpf}}(\hat{a}^\dagger - \hat{a})$ (zpf stands for zero point fluctuation), $\hat{a}^\dagger$ and $\hat{a}$ are the bosonic creation and annihilation operators of the harmonic oscillator, such that $\left[\hat{\phi}, \hat{N}\right] = i$, $[\hat{a}, \hat{a}^\dagger] = 1$, and $\phi_{\text{zpf}} = \left(\frac{2E_C}{c_2 E_J}\right)^{1/4}$ where $c_n = \frac{1}{n!}\frac{\partial^n U}{\partial \phi}\big|_{\phi = \tilde{\phi}_{\min}}$ [54, 56, 57], the total Hamiltonian takes the form:

$$\hat{H}/\hbar = \omega_r \hat{a}^\dagger \hat{a} + \sum_{n=3}^{\infty} g_n (\hat{a} + \hat{a}^\dagger)^n \tag{C3}$$

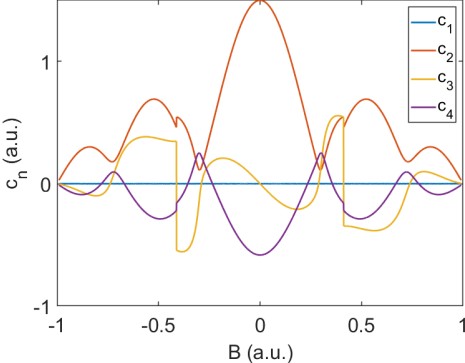

FIG. 10. CPR with large second and third harmonic components.

where $\omega_r$ is the resonant frequency of the simple harmonic oscillator, shown in Fig. (11). We truncate the series to fourth order non-linearity under the condition $\phi_{\text{zpf}} \ll 1$, hence,

$$\hat{H}_{\text{nl}} = g_3(\hat{a} + \hat{a}^\dagger)^3 + g_4(\hat{a} + \hat{a}^\dagger)^4 \tag{C4}$$

The nonlinear parameters $g_n$ are:

$$g_3 = \frac{1}{6}p^2\frac{c_3}{c_2}\sqrt{E_C\omega_r} \tag{C5}$$

FIG. 11. Resonant frequency $\omega_r$ with field $B$.

## Appendix D: Amplifier performance estimation

$$g_4 = \frac{1}{12}p^3\left(c_4 - \frac{3c_3^2}{c_2}(1-p)\right)\frac{E_C}{c_2} \tag{C6}$$

where $p = \frac{\zeta_J}{c_2+\zeta_J}$, $\zeta_J = L_J/L$, $L_J$ and $L$ are the Josephson inductance and linear inductance of the circuit respectively.

At the sweet spot, $|g_3| \approx 33$ MHz, $g_4 \approx 12$ Hz and $\omega_r \approx 3.98$ GHz. Using these parameters, we estimate the power gain of $\phi_0$-junction Josephson parametric amplifier for the three-wave mixing scheme. Using Eq. (6) and assuming $|\alpha_s| \approx |\alpha_i| = 0.01$ for low signal power and $\kappa = 400$ MHz, we find an optimum pump strength for achieving maximum gain shown in Fig. 12(a). In Fig. 12(b), we show gain vs. signal strength $|\alpha_s|$ with maximum gain at $\omega_s = \omega_p/2$, and a gain of 20 dB achieved over a bandwidth of 40 MHz. These values are an upper estimate on performance, which in real devices may be limited by additional factors that are not included in this calculation.

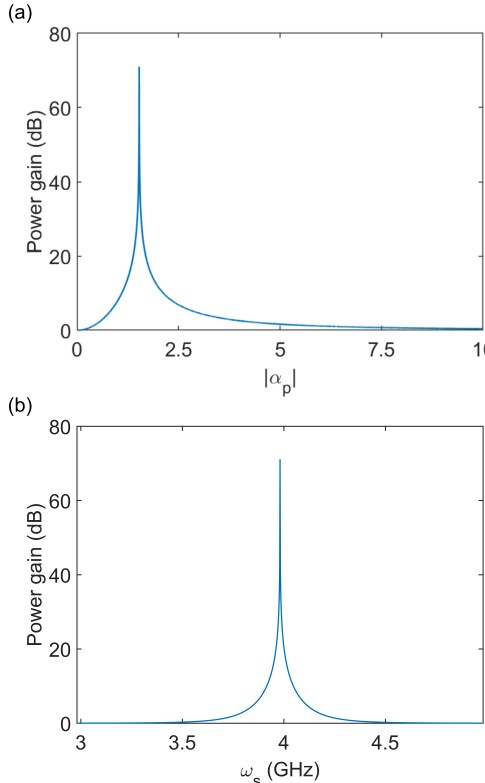

FIG. 12. (a) Power gain vs. pump amplitude. (b) Power gain vs. signal frequency at optimum pump amplitude.