# Peer review of "Predicted third-order sweet spots for phi-junction Josephson parametric amplifiers"

_SciPost Physics_

## Round 1 · Referee Report · Anonymous (Referee 1) · 2025-9-17

Disclosure of Generative AI use

The referee discloses that the following generative AI tools have been used in the preparation of this report:

grammar checking, microsoft copilot

Report

In the manuscript, the authors present a proposal for a resonator-type Josephson parametric amplifier based on a hybrid superconductor–semiconductor nanowire. This proposal builds upon their previous work published in SciPost Phys. 18, 013 (2025), where the current-phase relation (CPR) of a hybrid Josephson junction lacking inversion symmetry was demonstrated.
The design relies on the nonsinusoidal CPR of hybrid nanowire Josephson junctions (JJs). Although Sn/InSb or Sn/InAs junctions are likely not in the short-junction limit, their CPRs are nevertheless expected to contain higher-order harmonics. This is a reasonable assumption, and the ratio between the amplitudes of the first and second harmonics, I1/I2≈4, is plausible. However, the most limiting factor remains the barrier between the superconductor and semiconductor, as discussed in Phys. Rev. B 71, 052506 (2005).
As correctly stated in the manuscript, the mere presence of higher-order CPR terms is not sufficient to realize a single junction that dominates three-wave mixing. However, some statements such as “fourth-order nonlinear term of the inductive potential” and “A common device with third-order nonlinearity is the Superconducting Nonlinear Asymmetric Inductive eLement (SNAIL)” may be confusing to readers.
I suggest defining the nonlinear Josephson inductance LJ  early in the manuscript and expressing LJ(ϕ) for the CPR given in Eq. (1). It may even be possible to derive a condition under which the even nonlinearity of LJ  vanishes—a “sweet spot.” Additionally, the Josephson potential and Hamiltonian could be introduced, but in that context, I recommend using the terminology “higher-order anharmonic terms of the potential.”
The weakest point of the proposal is the rather complex magnetic field tuning required to bring the JJ to the sweet spot. It is unclear why electric field tuning has not been explored. Since the sweet spot should also depend on the ratio I2/I1 , which could be gate-voltage dependent, one could potentially use micromagnets and tune the JJ via gate voltage instead.
While the idea is promising, several aspects of the manuscript could benefit from clarification and refinement.
 
In the introduction section, the formulation "By way of assessment of device characteristics from key metrics such as power gain, 1 dB compression power and operable frequency bandwidth, Josephson junctions based on Sn as superconductor offer the critical current levels (100s of nA) that are compatible with SNAIL operation." is misleading, as it suggests that Sn-based Josephson junctions have already been benchmarked in terms of amplifier metrics (gain, compression power, bandwidth), comparable to SNAIL performance. In reality, the only parameter assessed here is the critical current, which indeed falls in the suitable range for SNAIL operation, but no amplifier-level comparison was carried out.
 
In the caption of Fig. 1, fitting parameters n_n1  and n_n2  are introduced for harmonic amplitudes I_n , extracted from numerical simulations. These parameters are not consistently used in the main text, and their values are not listed. Moreover, in Eq. (2), n_11  and n_21  are replaced by α and β, while n_12  and n_22  appear redundant. Since the magnetic field dependence of the critical current amplitudes is already normalized to the critical field Bc , the critical current should be treated as a fitting parameter (possibly with distinct values B_nc ), and n_n2  should be omitted.
 
The manuscript states that, to estimate parameters leading to optimal nonlinearity (Kerr-free operation with strong third-order nonlinearity), an optimization was performed over two variables, a and c. However, as correctly noted in the text, parameter a does not affect the shape of the potential well around the minimum. This is also clearly illustrated in Figs. 6(a) and 6(b). From Eq. (3), it follows directly that a corresponds only to a global phase shift and therefore has no effect on the local nonlinearity. Given this, it is puzzling that the optimization yields a precise value for a, as reported in the caption of Fig. 2.
 
On the other hand, parameter c (proportionality between mutual harmonic phase shift d_12 and the magentic field) plays a crucial role in determining the nonlinearity. In the manuscript, however, its origin and behavior are only briefly mentioned. In Ref. 46, it is stated, that d_12 arises as a consequence of spin–orbit interaction in the numerical simulations. The manuscript would benefit from an expanded discussion of the role and origin of d_12, including how its presence can be ensured and how much experimental control is available. Since the optimization of the nonlinearity relies critically on this parameter, clarifying its microscopic origin and practical tunability would be useful.
 
 
 
Figure 12 in Appendix D is mistakenly described: “In Fig. 12(b), we show gain vs. signal strength ∣αs∣ with maximum gain at ωs=ωp/2, and a gain of 20 dB achieved over a bandwidth of 40 MHz.” In fact, the figure shows gain versus signal frequency. Nevertheless, a figure displaying gain versus signal strength would be of greater interest, as the use of the sweet spot is motivated by improvements in dynamic range. I would expect the 1 dB compression point to be estimated in order to demonstrate the viability of the optimization.
 
The manuscript presents an interesting proposal, but it requires revisions before it can be considered for publication. The authors should address the points above to improve clarity, consistency, and scientific rigor.

Attachment

Recommendation

Ask for minor revision

---

## Round 1 · Referee Report · Jean-Damien Pillet (Referee 2) · 2025-9-19

Disclosure of Generative AI use

The referee discloses that the following generative AI tools have been used in the preparation of this report:

Improve the quality of english grammar.

Strengths

  1. Extends a very interesting conceptual idea from Ref. 42, while providing a more quantitative treatment.
  2. Offers a promising and stimulating direction for realizing a parametric amplifier with a hybrid superconducting circuit.
  3. Clearly demonstrates the existence of “sweet spots” for optimal device operation.
  4. Exploits the electron spin–orbit interaction to generate a three-wave mixing term for microwave photons.

Weaknesses

  1. Some assumptions could be better supported in the manuscript, in particular the linear magnetic-field dependence of the phase offset of the second harmonic, $\delta_{12}$, which is a crucial ingredient for the device operation.
  2. Many quantities are given in arbitrary units. While the chosen values appear consistent and credible, it is unfortunate that physical units are not provided, as they would help the reader better assess feasibility.
  3. The novelty is relative, since Ref. 42 already proposed using Josephson junctions with strong spin–orbit coupling to achieve Kerr-free three-wave mixing.
  4. The micromagnet strategy, although clearly illustrated with a figure and supported by good arguments, remains relatively underdeveloped and somewhat superficial.
  5. Most of the work essentially consists of a Taylor expansion of a phenomenological expression for the Josephson potential, combined with analytical formulas already established in the SNAIL literature. There is no strong new theoretical results.

Report

First of all, I would like to apologize to the authors and the editor I could not offer to send my report earlier; personal and professional obligations prevented me from doing so.

Please, find my report below.

The manuscript by T. Reza and S. Frolov presents an original strategy to realize three-wave mixing, and ultimately Kerr-free parametric amplification, using a hybrid superconducting device based on a semiconductor nanowire with strong spin–orbit coupling. In the presence of a magnetic field, both inversion and time-reversal symmetries are broken, which allows cubic $\varphi^3$ terms to appear in the Josephson potential.

I find this an excellent paper and recommend its publication even in its current form. It presents a clear and convincing case for a novel and promising research direction. The reported values of the nonlinear coefficients $g_3$ and $g_4$ are particularly encouraging and suggest excellent potential performance for this amplifier, which could offer a credible alternative to the SNAIL, more compact and potentially more resilient to magnetic fields.

As is customary, I will nevertheless point out a few potential improvements that the authors may wish to address to strengthen the manuscript:

  1. Phenomenological current–phase relation (Eq. 2): The central equation is a phenomenological CPR that explicitly introduces magnetic-field dependence in both the critical current components and the phase offsets of the first and second harmonics, $\varphi_0$ and $\delta_{12}$. This CPR is justified by KWANT simulations (not shown here, but presented in Ref. 46 as I understand). The authors assume linear dependence of both offsets on the magnetic field. The linearity of $\varphi_0$ is well established (see, e.g., Nazarov, Anomalous Josephson effect induced by spin–orbit interaction and Zeeman effect in semiconductor nanowires), but explicit references supporting the linearity of $\delta_{12}$ are lacking. In Ref. 46, however, one can indeed infer linear growth of $\delta_{12}$ with field (Fig. 23c, shaded region). I suggest that the authors make this justification clearer, perhaps by reproducing that figure in an appendix, to guide the reader.

  2. Use of arbitrary units: Many figure axes are given in arbitrary units. While the manuscript provides enough information for conversion, it would be clearer and more convincing to plot the results directly in physical units, and to explicitly report the values obtained experimentally (references are already provided but without explicit numbers). This would make it immediately apparent that the chosen parameters, in particular $a$ and $c$, are realistic. Moreover, the current practice leads to inconsistent notation: sometimes $B$ is expressed in mT, sometimes normalized as $B_c = 1$, which reduces readability.

  3. Connection of the nanowire to the ground: In Fig. 4a, the right lead of the nanowire should be connected to the ground. It is presently floating.

  4. Power gain: The figure 12 in the appendix shows gain reaching 70 dB. This seems unrealistic. Do the authors believe they are showing a reasonable operating point ?

In conclusion, this is a highly interesting and valuable contribution. I warmly congratulate the authors on their work and recommend its publication.

Recommendation

Publish (meets expectations and criteria for this Journal)

---

## Editorial Decision

awaiting_resubmission